# RRGA-Net: Robust Point Cloud Registration Based on Graph Convolutional Attention

**DOI:** 10.3390/s23249651

**Published:** 2023-12-06

**Authors:** Jian Qian, Dewen Tang

**Affiliations:** School of Mechanical Engineering, University of South China, Hengyang 421001, China; 20212006210312@stu.usc.edu.cn

**Keywords:** point cloud registration, deep learning, attention mechanism

## Abstract

The problem of registering point clouds in scenarios with low overlap is explored in this study. Previous methodologies depended on having a sufficient number of repeatable keypoints to extract correspondences, making them less effective in partially overlapping environments. In this paper, a novel learning network is proposed to optimize correspondences in sparse keypoints. Firstly, a multi-layer channel sampling mechanism is suggested to enhance the information in point clouds, and keypoints were filtered and fused at multi-layer resolutions to form patches through feature weight filtering. Moreover, a template matching module is devised, comprising a self-attention mapping convolutional neural network and a cross-attention network. This module aims to match contextual features and refine the correspondence in overlapping areas of patches, ultimately enhancing correspondence accuracy. Experimental results demonstrate the robustness of our model across various datasets, including ModelNet40, 3DMatch, 3DLoMatch, and KITTI. Notably, our method excels in low-overlap scenarios, showcasing superior performance.

## 1. Introduction

Point cloud registration is a crucial task within computer vision and robotics, frequently applied in significant domains like autonomous driving [1], 3D reconstruction [2], and simultaneous localization and mapping (SLAM) [3]. In recent years, with the development of point cloud processing technology and deep learning technology, point cloud coding algorithms such as [4,5] have further optimized the processing of large-scale point cloud data and improved the accuracy of point cloud registration in large-scale scenes. However, registration in low-overlap environments remains a challenging task.

A classical point cloud registration method is Iterative Nearest Point (ICP) [6]. It starts with an initial transform guess, and then iteratively updates the transform matrix to minimize the distance between corresponding points until convergence or a certain number of iterations is reached. The disadvantage is that it is sensitive to initial transformations and local minima. The Fast Global Registration (FGR) [7] algorithm addresses the drawbacks of the ICP algorithm through global alignment, but is still prone to failure in noisy environments.

Deep learning-based methods can be divided into three categories: the first category [8,9] is based on the global features of the point cloud, which is treated as a whole to regress the transformation parameters. Although this type of method has good robustness to noise, it is not effective for the registration task of partially overlapping point cloud. The second class of methods [10,11] based on correspondence learning form correspondences by means of high-dimensional features of the points and iteratively minimize the feature distances to optimize the pose. This type of approach extracts the correspondences of the points, so it is more robust in partially overlapping point cloud registration, but it is still susceptible to noise interference. The last class [12,13,14] uses a two-stage approach to the point cloud correspondences, where they first learn the local descriptors of the downsampled keypoints for matching, and then use a pose estimator to recover the relative transformations. Their strategy allows them to achieve state-of-the-art performance in the dataset, but the downsampling method they use inherently introduces sparsity point cloud features, which reduces the reproducibility of the correspondences and thus loses its advantage in low overlap regions.

A recently discovered approach [15] bypasses direct keypoint detection. It mitigates the limitations of keypoint detection by employing a coarse-to-fine strategy similar to the two-dimensional correspondence approach [16,17] to address the problem of keypoint sparsity. It achieves correspondence extraction by employing a superpoint patch-to-point path. Essentially, the method extracts superpoint features from the original point cloud, merges them into superpoint patches for matching, and then extends the matched correspondences to include dense points. The advantage of the method is that it transforms the strict point matching requirement into a more relaxed patch overlapping environment. This shift effectively reduces the requirement for a large number of repeatable keypoints. However, it also emphasizes the importance of keypoint reliability. Furthermore, while this approach reduces the need for a large number of keypoints, its sparsity remains unchanged. Therefore, in this paper, more emphasis is placed on compensating for this inherent sparsity by capturing contextual features.

Taking inspiration from references [15,18], our initial focus lies in optimizing the enhancement of the patch context function. With the objective in mind, a matching module based on a graph convolutional neural network was designed, the core of which is constituted by two modules: the self-attention graph convolutional neural network and the cross-attention module. Graph convolutional networks focus on different points when processing point cloud data and dynamically adjust weights based on the relationship between them. This adaptability enables the model to better capture local and global features in point cloud, thereby improving the modeling ability of template points. Secondly, the cross-attention module enables the model to effectively capture the information interaction in point cloud between different channels by introducing cross-channel correlation. This helps integrate multi-channel information to more fully understand feature relationships in point cloud. Through this cross-channel interaction, the model is better able to adapt to complex structures in point cloud data. In addition, previous methods use grid-sampled superpoints as nodes and divide patches through a point-to-node grouping strategy. Since the grid-sampled superpoints are sparse and loose, the local neighborhoods between point cloud pairs are inconsistent, which adversely affects subsequent point matching. To this end, it is recommended to extract points characterized by a more uniform visual field distribution and high repeatability as nodes. The utilization of the feature pyramid effect is suggested for scoring nodes across various receptive fields. Additionally, incorporating multi-level point cloud sampling is advised during the process of patch fusion. During the sampling of point cloud at different levels, the density of sampled points is regulated through non-maximum suppression. Utilizing multi-channel sampling and point matching comparison allows for the acquisition of more comprehensive point cloud information, resulting in improved correspondence with template points.

## 2. Related Work

### 2.1. Traditional Point Cloud Registration Methods

The Iterative Closest Point (ICP) [6] algorithm has retained its practical importance since it became established. Its straightforward logic and ease of implementation have solidified its position as a staple method for aligning rigid-body transformations. It shows strong convergence when the initial deviation is relatively small. However, its accuracy decreases when the initial bias is large, resulting in local optima and sensitivity to noise. Consequently, an array of ICP-based variants [19,20,21] have emerged.These ICP-based variations expand upon the original concept to address its limitations. They offer enhanced flexibility by accommodating multiple constraints such as distance, geometry, and normals, leading to improved robustness against initial deviations. However, this enhancement often comes at the cost of increased computational complexity.

Among alternative strategies, feature-based registration methods like Fast Point Feature Histograms (FPFH) [22] and Signature of Histograms of Orientations (SHOT) [23] have been developed to adapt to complex environments and noise by extracting local features. The Random Sample Consensus (RANSAC) [24] divides the point cloud into random subsets for local registration, effectively eliminating outliers, albeit at the expense of computational time. Conversely, the Fast Global Registration (FGR) [7] transforms the non-convex problem into a convex one through a smoothing mechanism, enabling rapid global registration. However, this method exhibits heightened sensitivity to errors.

### 2.2. Deep Learning-Based Methods

PointNet [25] is the first deep learning model that can directly process raw point cloud data without converting the point cloud into voxels or meshes. Thus, PointNetLK [8] uses PointNet’s ability to extract global features from point cloud to achieve alignment via the LK optical flow method in classical image alignment. PCRNet [9], on the other hand, uses a multilayer perceptron (MLP) to solve the transformation parameters by treating the registration task as a regression problem after extracting features using PointNet. But PointNet, although it is easy to extract global features of the point cloud, loses the value of local features, so it is not robust to noisy and partially overlapping environments.

In contrast to approaches relying on PointNet, DCP [10] uses a Dynamic Graph Convolutional Neural Network (DGCNN) [26] to extract local features from the original point cloud. The rotation matrix and translation parameters are then computed by Singular Value Decomposition (SVD). RPMNet [27] introduces an auxiliary network to predict the optimal asymptotic annealing parameters to derive soft matches for point correspondences in integrating spatial coordinates and local geometric features. These approaches based on local correspondence learning solve part of the point cloud matching problem to some extent, but the network fails to converge when the rotation angle is too large. In the method of correlated feature point detection, D3Feat [14] utilizes a fully convolutional network architecture for joint dense detection and description. Predator [12] predicts dense overlap scores on top of jointly estimating significant scores and learning local descriptors, and analyses the confidence level of whether a point is located on an overlapping region. However, they show low robustness in low overlap scenarios. This is due to the fact that they still inherently rely on repeatable keypoints.

A facet-to-point correspondence strategy is employed, and the establishment of point cloud correspondence is carried out in a coarse-to-fine manner. The dependence on repeatable key points is reduced by composing patches. Meanwhile, the multi-channel sampling strategy has a denser set of correspondence points, while the attention mechanism-based graph convolutional neural network enriches the correspondence of template points by interacting with contextual features. The performance of various algorithms will be compared in Section 4.

## 3. Methods

Point cloud registration involves aligning point cloud data of an object from distinct viewpoints or sensors onto the same coordinate system. The objective is to fit multi-frame images, enhance visual perception to facilitate understanding of the environment, and provide assistance in subsequent tasks.

The central challenge of point cloud registration revolves around aligning two distinct point clouds, $P=\{p_{i}\in R^{3}|i=1,\ldots ,N\}$ and $Q=\{q_{i}\in R^{3}|i=1,\ldots ,M\}$, while optimizing their alignment within the shared coordinate system. This task can be represented through the subsequent model: how can we determine a transformation matrix, denoted as T={R,t}, which effectively reposition point cloud Q to attain optimal conformity with the spatial orientation of point cloud P? This issue can be effectively characterized as an optimization problem:(1)$$\underset{R,t}{\mathrm{argmin}}\sum_{i=1}^{N}\sum_{i=1}^{M}{∥Rp_{i}+t-q_{i}∥}_{2}^{2}$$
where R denotes the rotation matrix and t denotes the translation parameter. We estimate the alignment transformation by finding point correspondences.

The point cloud registration model is illustrated in Figure 1. KPConv and FPN are utilized simultaneously to downsample the input point cloud and extract features (refer to Section 3.1). The downsampled points of the three different levels of the three layers are also selected as the reference and feature points of the correspondences to be matched, respectively. The graph convolutional neural matching module is used to extract the correspondences of the feature points (Section 3.2). Subsequently, the point matching module employs these correspondences to extend the alignment from feature points to encompass the entire dense point cloud (Section 3.3). At last, the local-to-global alignment method estimates the transformation matrix.

### 3.1. Feature Extraction

The KPConv-FPN [28] technique is utilized to downsample the initial point cloud and obtain features at the individual point level. Multiple resolution levels are sampled from the original point cloud P. These layers of sampled point cloud exhibit progressively diminished resolution connections, necessitating a coarse-to-fine methodology to master correspondences. The points corresponding to the most rudimentary resolution, designated as $\hat{P}$, are regarded as the reference points to be aligned. Both multi-level transition points, denoted as $\bar{P}$, and dense points, denoted as $\tilde{P}$, are independently extracted, and their respective acquired features are labeled as ${\bar{F}}^{P}\in R^{|\bar{P}|\times \bar{d}}$ and ${\tilde{F}}^{P}\in R^{|\tilde{P}|\times \tilde{d}}$. *d* is the corresponding sampling scale, determined by the resolution factor. For each modal point, a point-to-node grouping strategy is employed, thereby constructing localized point patches around it. The features in $\bar{F}$ and $\tilde{F}$ are assigned to the nearest modal point, and the ensuing equation validates the assigned weights:(2)$$w=∥\tilde{p}-\hat{p}{∥}_{2}/{∥\bar{p}-\hat{p}∥}_{2}$$

Based on the weights, for different levels, the nearest points will be attributed to the modal points closest to them and the resulting patch is shown below:(3)$$G_{i}^{P}=\left\{\begin{array}{cc} i={\mathrm{argmin}}_{j}(∥\bar{p}-{\hat{p}}_{j}{∥}_{2}),{\hat{p}}_{j}\in \hat{P} & w>1,\\ i={\mathrm{argmin}}_{j}(∥\tilde{p}-{\hat{p}}_{j}{∥}_{2}),{\hat{p}}_{j}\in \hat{P} & w\leq 1, \end{array}\right.$$

### 3.2. Graph Convolutional Neural Network Matching Module

Obtaining a global field of view is critical in a variety of computer vision tasks. Therefore, we employ an attention mechanism that utilizes broader contextual information to augment global properties. This yields enhanced geometric differentiation within the acquired features, consequently mitigating pronounced matching ambiguities and a surplus of aberrant matches, particularly in scenarios characterized by limited overlap. Through the utilization of the cross-focusing mechanism, feature details from the point cloud can be adeptly interchanged and amalgamated, leading to the identification of pivotal points connected with regions of overlap. This innovative approach effectively solves the problem of redundant point accumulation while simplifying the process of selecting point sets during the alignment process. In the cross-focus module, the initial embedding contains features from both the source and target point cloud.

Before connecting the feature codes of the two inputs, a graph neural network (GNN) is first used to further aggregate and strengthen their contextual relationships, respectively. The point sets P or Q are connected into a graph within the Euclidean space through the employment of the K-Nearest Neighbors (K-NN) algorithm. Subsequently, utilizing K-NN searches based on coordinates, the features are linked to centroid features.
(4)$$f_{i}=cat\left(x_{i}^{n},x_{j}^{n}-x_{i}^{n}\right)$$

In the aforementioned equation, $x^{n}$ signifies the feature encoding corresponding to the point set P, while “cat” denotes concatenation.
(5)$$h_{i}=\mathrm{LeakyRLU}\left(\mathrm{norm}\left(\mathrm{conv}\left(f_{i}\right)\right)\right)$$
(6)$$x_{i}^{n+1}=\max_{(i,j)\in \varepsilon }h_{i}\left(f_{i}\right)$$
where $h_{i}$ denotes the linear layer, norm denotes the activation function, max denotes the elemental channel maximum pooling layer, and (i,j)∈ε denotes the two edges of the graph.
(7)$$x_{i}^{self-GNN}=h_{i}\left(cat\left(x_{i}^{0},x_{i}^{1},x_{i}^{2}\right)\right)$$

Cross-attention stands as a prototypical module within point cloud registration tasks, fostering the exchange of features between two input point clouds. We apply self-attention processing to the three-channel point cloud data, combining the resultant data with convolved point cloud information, and subsequently activate the aggregated point cloud using a Multi-Layer Perceptron (MLP). The computation of the Cross-Attention (CA) module is detailed as follows:(8)$$m_{i}=\mathrm{att}\left(x_{i},x_{j},x_{j}\right)$$

Here, att denotes Multi Head Attention. $x_{i}=x_{i}^{self-GNN}$, $x_{j}=x_{j}^{self-GNN}$.
(9)$$X_{i}^{CA}=x_{i}^{self-GNN}+\mathrm{MLP}\left(\mathrm{cat}\left(x_{i}^{self-GNN},m_{i}\right)\right)$$

In the sampling of correspondences, considering patch correspondences at different levels helps to obtain more robust point correspondences in the point matching stage. However, due to the sparse and loose nature of block matching, many correct correspondences are often overlooked in the screening process. Our proposed multi-channel convolutional network can complement more effective point correspondences for accurate point cloud registration.

### 3.3. Point Matching Module

After obtaining the template point correspondence, the dense point correspondence will be derived based on the patch correspondence. Subsequently, the local-to-global registration (LGR) mechanism derives candidate matrices from the point correspondences engendered by each pairing of matching patches. From these candidates, the globally optimal transformation matrix is selected. Pertaining to the point-level, our approach exclusively employs localized point features gleaned from the backbone network. The underlying principle is that, upon resolving global ambiguity through template point matching, point-level matching is predominantly influenced by the proximity of the matched points. This strategic design enhances the overall robustness of the process.

For each template point correspondence, the optimal transport layer is employed to derive localized dense point correspondences between the point clouds. The process begins by calculating the cost matrix:(10)$$C_{i}=F_{x_{i}}^{P}{\left(F_{y_{i}}^{Q}\right)}^{T}/\sqrt{\tilde{d}},$$

Following this, the cost matrix undergoes expansion through the addition of new rows and columns, infused with learnable bin parameters. Subsequently, the Sinkhorn algorithm is employed to compute the soft assignment matrix. This matrix is then reinstated to its original form by discarding the last row and column. This resultant matrix serves as a confidence measure for prospective matches. Point correspondences are subsequently culled through mutual top-k selection, whereby a point match is affirmed if it falls within the k largest entries within its respective row and column. The point correspondences computed from each template point match are then collected together to form the final global dense point correspondence: $C={⋃}_{i=1}^{N_{c}}C_{i}$.

### 3.4. Loss Functions

The loss functions of the Graph Convolutional Neural Matching Module and the Point Matching Module are divided into the following two points.

A metric learning approach is chosen to cultivate a feature space that facilitates the assessment of the similarity of samples. This approach is tailored to more effectively evaluate the matching interrelation between patches, facilitating the convergence of matches and divergence of mismatches. We meticulously select patches within P, ensuring each belongs to a group of anchor point patches denoted as N, where a positive patch in Q is present. Pairs of patches are categorized as positive if they display a minimum of 10% overlap, and conversely as negative if they lack overlap. All other pairs are omitted from consideration. For each anchor patch $G_{i}\in N$, a corresponding loss takes on the following format:(11)$$L_{c}=\sum_{G_{i}^{P}\in N}{\left[{∥f(x_{i}^{a})-f(x_{i}^{p})∥}_{2}^{2}-{∥f\left(x_{i}^{a}\right)-f\left(x_{i}^{n}\right)∥}_{2}^{2}+\alpha \right]}_{+}$$
where $x_{i}^{a}$ symbolizes the feature representation of the anchor, while $x_{i}^{p}$ signifies the feature representation of the positive example that aligns with the anchor patch. Correspondingly, $x_{i}^{n}$ stands for the feature representation of the negative example, which does not align with the anchor patch. The parameter α functions as a constant threshold, integral to guaranteeing that the disparity in distance between the positive and negative examples remains surpassing a predefined threshold. The function ${\left[z\right]}_{+}=\max\left(z,0\right)$ corresponds to the Rectified Linear Unit (ReLU) function. By cultivating a suitable feature space and employing these strategies, we optimize the discernment of matching relationships among patches, thereby enhancing the precision of point cloud registration.

The correspondence relationship of the real point set is sparser than that of the downsampled template points. The correspondence matrix Z in point matching is classified using a negative log-likelihood loss.

During training, true point correspondences $\hat{C}i$ are randomly sampled. For each $\hat{C}i$, a set of true point correspondences M is extracted using the matching radius *r*. The set of unmatched points in the two patches is denoted as $I_{i}$ and $J_{i}$. The individual point matching loss of ${\hat{C}}_{i}$ is computed as:(12)$$L_{p}=-\sum_{(x,y)\in M}\log{\bar{z}}_{x,y}^{i}-\sum_{x\in I}\log{\bar{z}}_{x,m_{i}+1}^{i}-\sum_{y\in J}\log{\bar{z}}_{n_{i}+1,y}^{i}$$

The final loss function X consists of three loss functions together: $L=L_{c}+L_{m}+L_{p}$, where $L_{m}$ is different from $L_{p}$. The intermediate layer $\bar{P}$ is used as the real point set. This approach establishes a link between multiple levels of point correspondences, and by exploiting these multiple levels of point correspondences, our method can compute point cloud feature parameters from a comprehensive perspective. This strategy not only improves the accuracy of point cloud alignment, but also enriches the representation learning process by exploiting the hierarchical structure inherent in the data.

## 4. Results

This section is dedicated to the experimental validation and performance comparison of our proposed method. The efficacy of the model is meticulously assessed through comprehensive experimentation. To establish a robust basis for evaluation, comparisons are conducted against several established methods, namely ICP, FGR, PointNetLK, DCP, and RPMNet. For the evaluation process, the ModelNet40 dataset [29] is employed as a benchmark. Through testing and analysis, distinctive advantages offered by our model in contrast to these existing methodologies are elucidated. Furthermore, to ascertain the generalizability of our approach in real-world scenarios, the evaluation is extended to encompass actual data. Engagement with the 3DMatch [30], 3DLoMatch [31], and KITTI [32] datasets is carried out to test the adaptability and reliability of our model within practical contexts.

### 4.1. ModelNet40

Our algorithm undergoes a thorough evaluation process on the ModelNet40 dataset, which encompasses computer-aided design (CAD) models representing 40 diverse classes of human-made objects. The evaluation strategy involves training on a set of 9756 models and testing on a separate collection of 2555 models. In alignment with the experimental framework established by RPMNet, adherence to specific guidelines is maintained. For each given shape, 1024 points are selected to constitute a point cloud. Additionally, an element of randomness is introduced into the evaluation process. Specifically, three Euler angles per point cloud are generated, each within the range of [0, 90°]. Furthermore, translations are introduced within the range of [−0.5, 0.5]. The original and target point clouds are distinguished in red and green.

A consistent metric framework is adopted, aligning with the assessment criteria employed by RPMNet [11] to evaluate the performance of our algorithm. This approach ensures comparability with previous research and underscores the reliability of our results. In this metric framework, the evaluation of alignment is performed by calculating the average isotropic rotation and translation errors, along with the average absolute errors of the Euler angles and translation vectors. If the overlapping regions of the two point clouds are identical, then all error parameters should be close to zero.

The performance of the algorithm is thoroughly evaluated across various point cloud scenarios, including clean point cloud, environments with noise, and instances of partially visible point cloud. The experimental outcomes are graphically presented in Figure 2 and Table 1, Table 2 and Table 3. Since some algorithms do not have reliable open source implementations, some data come from their papers.

From Figure 2, it can be concluded that traditional methods such as ICP are susceptible to initialization, which is particularly obvious when the rotation angle is large. On the other hand, the efficacy of FGR is weakened in noisy environments because FPFH is sensitive to the noise problem under different point cloud conditions. In contrast, PointNetLK performs well in noisy environments, but still faces challenges in partially visible data. The reason for this phenomenon is that global feature methods emphasize the overall features of the point cloud rather than the specific local features of individual points. GeoTransformer works well in clean and noisy point clouds. Even in cases involving partially visible noise, superior registration results are observed for point clouds with simple structures (d). However, in the case of point cloud (e), our algorithm outperforms GeoTransformer. This is because injecting geometric information can improve performance, but the estimation method based on the geometric transformer does not rely on a stable estimator like RANSAC, which increases the difficulty in the estimation of actual super points, and GNN contains the transformation of the kNN graph, invariance, and better performance in transformation estimation. Therefore, our method improves the registration effect more significantly.

The tabular data in Table 1, Table 2 and Table 3 is further analyzed. Table 1 shows that FGR performs better than us in clean data, and then its performance in noisy data reflects the previous inference. The following focuses on comparing some visible noise point cloud data in Table 3. The PointNetLK algorithm, which performs well in Table 2, meets its Waterloo, and the DCP also suffers more in the absence of point clouds. Our algorithm still maintains a relatively excellent score, while still improving compared to the data of GeoTransformer. In terms of registration efficiency, ICP boasts the simplest algorithm structure. Given the small point cloud base in this experiment, it highlights the superiority of geometric algorithms. Neither FGR nor ICP utilizes the GPU, resulting in FGR’s efficiency not being significantly enhanced after adding normal vector calculations. DCP adopts an end-to-end design model, eliminating the disadvantages of multi-stage calculation iterations seen in other algorithms, and it performs exceptionally well in computational efficiency due to GPU utilization. However, as the size of the point cloud increases, the performance of the end-to-end algorithm will experience nonlinear decline. GeoTransformer, utilizing geometric information to improve registration speed, achieves faster transformation estimation by omitting RANSAC. In contrast, our method utilizes multi-level feature extraction which, although it reduces part of the registration speed, increases the accuracy of feature extraction. Furthermore, the introduction of template points through hybrid sampling enhances the effectiveness of plane segmentation and feature matching.

### 4.2. Indoor Benchmarks: 3DMatch and 3DLoMatch

The point cloud data of the real environment is more complex than ModelNet40. The larger number of point cloud and lower overlap area will make many algorithms effective on ModelNet lose their advantages. The robustness of our algorithm is assessed in real environments with low overlap, specifically on the 3DMatch and 3DLoMatch datasets.

The 3DMatch dataset comprises a total of 62 scenes, distributed for training (46 scenes), validation (8 scenes), and testing (8 scenes) purposes. The 3DLoMatch dataset is a more challenging dataset derived from 3DMatch. In the original 3DMatch dataset, only point cloud pairs with an overlapping rate greater than 30% are employed for testing. In contrast, the testing set of 3DLomatch includes point cloud pairs with an overlapping rate ranging between 10% and 30%. Preprocessed training data are utilized, and its performance is evaluated using the established 3DMatch protocol.

In line with previous assessments, the performance of our algorithm is measured using three distinct metrics:Interior Point Ratio (IR): this metric quantifies the proportion of hypothetical correspondences with residuals falling below a predetermined threshold (e.g., 0.1 m) under the ground truth transformation;Feature Matching Recall (FMR): FMR denotes the fraction of point cloud pairs wherein the interior point ratio surpasses a specified threshold (e.g., 5%);Matching Recall (RR): RR involves evaluating the fraction of point cloud pairs exhibiting transformation errors below a given threshold (e.g., Root Mean Square Error < 0.2 m).

Experiments were performed identically on data from FCGF [33], D3feat, Predator, Cofinet and GeoTransformer (data obtained from the paper). As can be seen from Table 4, our model achieves the best performance in all three indicators. In 3DLoMatch (Table 5), compared to GeoTransformer, the FMR indicator is slightly insufficient. This is because when the point cloud overlap rate is too low, although multi-level sampling increases the number of point cloud pairs, the number of tasks with high confidence in the total registration tasks will decrease. Figure 3a,b represents the registration results of 3DMatch, and Figure 3c–e represents the registration results of 3DLoMatch. The algorithm achieves good registration results in both data sets.

### 4.3. Outdoor Benchmark: KITTI Odometry

The KITTI odometry dataset encompasses 11 sequences capturing diverse outdoor driving scenarios, all of which are captured using LiDAR technology. Our utilization of this dataset is distributed as follows: sequences 0 to 5 are designated for training purposes, sequences 6 and 7 serve as validation sets, and sequences 8 to 10 constitute the testing data. In alignment with established practices in prior research, we adhere to the stipulation that only point cloud pairs separated by a minimum distance of 10 m are considered for evaluation.

Consistent with established practices in earlier studies, our performance evaluation hinges on three critical metrics:Relative Rotation Error (RRE): this metric quantifies the geodesic distance between the rotated matrix derived from our method and the corresponding ground truth rotated matrix;Relative Translation Error (RTE): RTE computes the Euclidean distance between the rotated matrices and the ground truth translation vectors;Recall to Alignment (RR): RR is a comprehensive metric reflecting the fraction of point-cloud pairs wherein both the RRE and RTE fall below specified thresholds (e.g., RRE < 5° and RTE < 2 m).

As shown in Table 6, our model is compared with [12,14,15,18,33]. In comparison to the real environment on the ground, we ensured similar displacement errors and rotation errors. Compared to other models, our metrics do not open a large gap, but still demonstrate the good generalizability of our model in outdoor environments. The registration effect is shown in the Figure 4.

### 4.4. Ablation Experiment

To illustrate the influence of each component on network performance, an ablation study is conducted in this section. Various modules within the network are systematically added and removed, allowing for an evaluation of their respective contributions to the final matching performance. Experiments are conducted on partially visible point cloud with noise. For easier comparison of results, we selected relative rotation error (RRE), relative translation error (RTE), and root mean square error (RMSE) as measurement standards. Experimental results (Table 7) show that a single graph convolution module can improve some accuracy, but there is still a gap with Transformer. After adding the cross-attention mechanism, our module indicators are already better than Transformer. The addition of Multi-channel further increased the rotation error and translation error by 7.3% and 5.8%. In addition, experimental results prove that our improved method achieves performance improvement compared to the baseline model and has good versatility.

## 5. Conclusions

This paper proposes a novel network to solve the problem of point cloud registration in low-overlap environments. Compared with previous work, our model uses multi-layer patches to enrich correspondences and can still extract reliable correspondences from disordered point clouds in the environment of sparse keypoints. In addition, the template point matching module enhances the contextual features of patches through graph convolutional neural networks and multiple sub-attention mechanisms, guiding the model to match nodes with nearby regions and narrowing the search space for subsequent refinement. Experiments on multiple datasets show that our proposed method is very robust and still has high general-purpose capabilities on outdoor datasets.

## Figures and Tables

**Figure 1 sensors-23-09651-f001:**
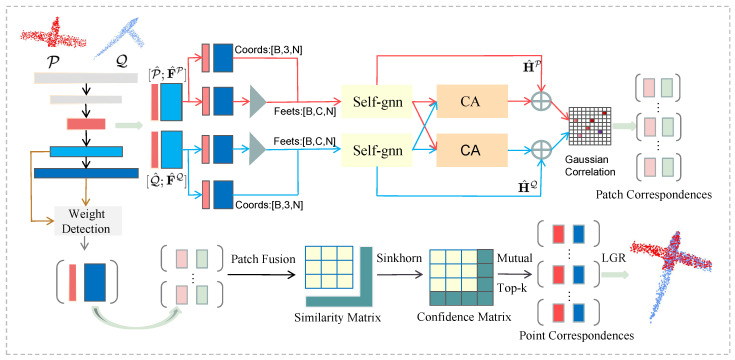
Given two partially overlapping point clouds P and Q, we first adopt KPConv-FPN to learn point cloud features at different levels. ${\bar{F}}^{P}$ or ${\bar{F}}^{Q}$ are connected to the centroid features through the K-NN algorithm and then imported into the graph convolutional neural network matching module. Their contextual features are further aggregated and enhanced through the self-attention graph convolution network (Self-gnn) and cross-attention (CA) block. Subsequently, the patch correspondences is mapped to the real point set through the point matching module. Finally, a local-to-global registration method is used to calculate the transformation matrix.

**Figure 2 sensors-23-09651-f002:**
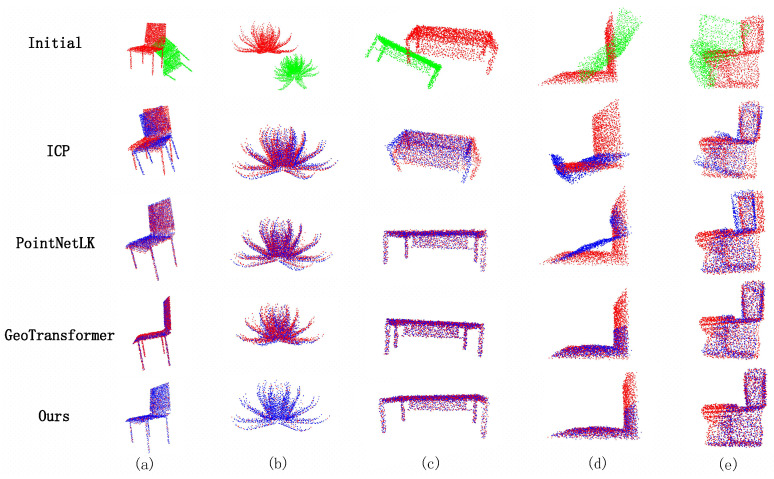
Examples of qualitative registration: (**a**,**b**) clean data, (**c**) noisy data, (**d**,**e**) partially visible noisy data.

**Figure 3 sensors-23-09651-f003:**
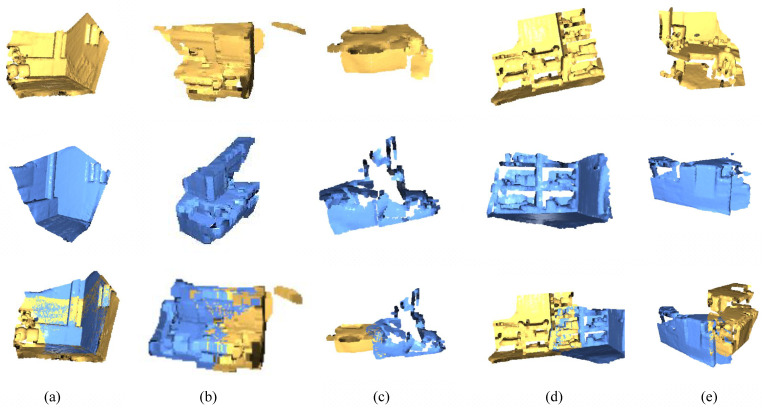
Results of 3DMatch and 3DLoMatch experiments. (**a**,**b**) are from 3DMatch, while (**c**–**e**) are from 3DLoMatch.

**Figure 4 sensors-23-09651-f004:**
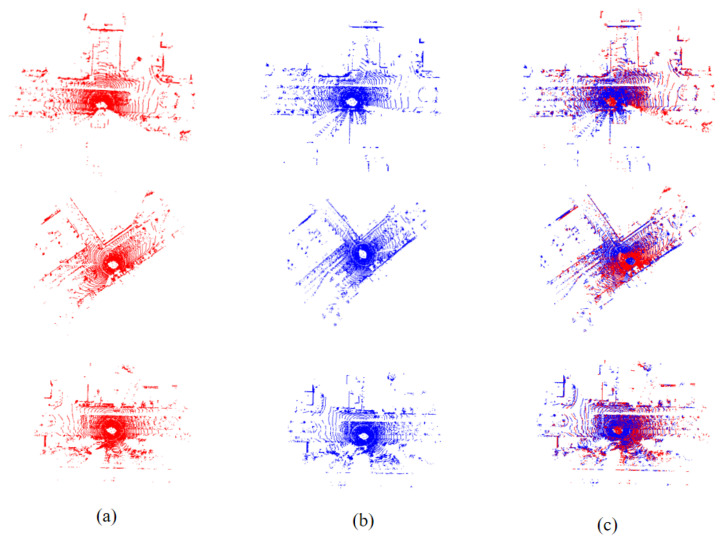
Results of KITTI experiments. (**a**,**b**) Point clouds collected using different radar positions and (**c**) is the result after registration.

**Table 1 sensors-23-09651-t001:** Performance on clean data.

Model	Isotropic R(°)	Isotropic t(m)	Anisotropic R(°)	Anisotropic t(m)	Time (s)
ICP	5.478	0.0765	11.443	0.1625	0.013
FGR	0.010	0.0001	0.022	0.0002	0.086
PointNetLK	0.418	0.0241	0.847	0.0054	0.157
DCP	2.074	0.0143	3.992	0.0292	0.009
PCR	2.691	0.0346	5.682	0.0735	0.059
GeoTransformer	0.072	0.0025	0.091	0.0023	0.023
Ours	0.034	0.0003	0.074	0.0005	0.052

**Table 2 sensors-23-09651-t002:** Performance on data with Gaussian noise.

Model	Isotropic R(°)	Isotropic t(m)	Anisotropic R(°)	Anisotropic t(m)	Time (s)
ICP	5.863	0.0823	12.145	0.1726	0.024
FGR	2.483	0.0325	4.274	0.0631	0.118
PointNetLK	1.528	0.0128	2.926	0.0262	0.214
DCP	4.528	0.0345	8.922	0.0707	0.020
PCR	2.943	0.0417	6.255	0.0804	0.122
GeoTransformer	1.156	0.0097	1.437	0.0213	0.045
Ours	0.525	0.0072	1.325	0.0127	0.083

**Table 3 sensors-23-09651-t003:** Performance on partially visible data with noise.

Model	Isotropic R(°)	Isotropic t(m)	Anisotropic R(°)	Anisotropic t(m)	Time (s)
ICP	13.719	0.132	27.250	0.280	0.017
FGR	19.266	0.090	30.834	0.192	0.124
PointNetLK	15.931	0.142	29.725	0.297	0.176
DCP	6.380	0.083	12.607	0.169	0.014
PCR	4.437	0.065	9.218	0.135	0.146
GeoTransformer	1.332	0.015	2.213	0.052	0.067
Ours	0.917	0.012	1.577	0.018	0.101

**Table 4 sensors-23-09651-t004:** Registration results on 3DMarch.

Model	IR (%)	FMR (%)	RR (%)
FCGF [33]	48.7	97.0	83.3
D3feat [14]	40.4	94.5	83.4
Predator [12]	57.1	96.5	90.6
CoFiNet [15]	51.9	98.1	88.4
GeoTransformer [18]	70.3	97.7	91.5
ours	72.5	98.5	93.0

**Table 5 sensors-23-09651-t005:** Registration results on 3DLoMarch.

Model	IR (%)	FMR (%)	RR (%)
FCGF [33]	17.2	74.2	38.2
D3feat [14]	14.0	67.0	46.9
Predator [12]	28.3	76.3	61.2
CoFiNet [15]	26.7	83.3	64.2
GeoTransformer [18]	43.3	88.1	74.0
ours	45.6	87.2	75.3

**Table 6 sensors-23-09651-t006:** Registration results on KITTI odometry.

Model	RRE (°)	RTE (m)	RR (%)
FCGF [33]	0.30	9.5	96.6
D3Feat [14]	0.30	7.2	99.8
Predator [12]	0.27	6.8	99.8
CoFiNet [15]	0.41	8.2	99.8
GeoTransformer [18]	0.24	6.8	99.8
ours	0.218	5.4	99.8

**Table 7 sensors-23-09651-t007:** Ablation experiments.

Baseline	Trans-Former	Self-gnn	CA Blocks	Multi-Channel	RRE (°)	RTE (m)	RMSE
√					2.154	0.033	0.026
√	√				1.577	0.018	0.017
√		√			1.723	0.029	0.021
√		√	√		1.554	0.017	0.016
√		√	√	√	1.44	0.016	0.015

## Data Availability

The data presented in this study are not currently publicly available but are available from the authors upon reasonable request.

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
