# Peer review of "RRGA-Net: Robust Point Cloud Registration Based on Graph Convolutional Attention"

_sensors, 2023, doi:10.3390/s23249651_

Round 1
Reviewer 1 Report
Comments and Suggestions for Authors
In this paper, the authors propose a novel learning network to optimize correspondences in sparse keypoints. Here are a few simple questions to ask.
1. Please further discuss the innovative points of the paper, rather than using existing algorithms to achieve point cloud registration.
2. There are a large number of non-standard first person discussions in the paper, please revise to a third person discussion.
3. Please add a comparison of time efficiency between different algorithms in the experimental section.
4. Is this algorithm suitable for large-scale point cloud registration?
Comments on the Quality of English Language
Minor editing of English language required
Reviewer 2 Report
Comments and Suggestions for Authors
The topic is relevant with the GeoTransformer and focuses on the low overlapping scenarios, but the experiments not sufficient to support the superior of the proposed method in current form.
The paper focused on the problem of registering point clouds in low overlapping scenarios, design a template matching module composed of a self-attention mapping convolutional neural network and a cross-attention network to match contextual features and refine the correspondence of patch overlapping areas to achieve better correspondence accuracy.
This paper proposes a novel network to solve the problem of point cloud registration in low-overlap environments. But the experiments not test on the low overlap dataset 3DLoMatch. Although the author claim "in real environments with low overlap on 3DMatch'', the results of IR,FMR,RR,is same as the Geotransformer test on 3DMatch, not the 3DLoMatch. In order to verified the effectiveness of the proposed algorithm, the author must be add experiments on the 3DLoMatch datasets.
The current version is not sufficient to support the conclusions. Detailed comparison with Algorithm GeoTransformer is required. Especially test on the 3DLoMatch dataset.
The references of this manuscript is lack of references from the past three years.
Tha tables and figures in 4.1 section, why not show the results compared with GeoTransformer in ModelNet40 dataset?
Comments on the Quality of English Language
The language expression of this paper is good.
Round 2
Reviewer 2 Report
Comments and Suggestions for Authors
The question was answered correctly, accept in current form.
Author Response
Thank you very much for your affirmation of our modifications, and thank you for your patience in responding.